**Cite this article:** Çoraman E, Dundarova H, Dietz C, Mayer F. 2020 Patterns of mtDNA introgression suggest population replacement in Palaearctic whiskered bat species. *R. Soc. Open Sci.* **7**: 191805.

evolution/genetics/ecology

Anatolia, the Balkans, the Caucasus, glacial refugia, hybridization, secondary contact

**Authors for correspondence:**
Emrah Çoraman
e-mail: coramane@gmail.com
Frieder Mayer
e-mail: frieder.mayer@mfn.berlin

†These authors contributed equally to this work.

# Patterns of mtDNA introgression suggest population replacement in Palaearctic whiskered bat species

Emrah Çoraman[1,2,3,†], Heliana Dundarova[4,†], Christian Dietz[5] and Frieder Mayer[2,6]

[1]Eurasia Institute of Earth Sciences, Department of Ecology and Evolution, Istanbul Technical University, Maslak, Istanbul 34469, Turkey
[2]Museum für Naturkunde, Leibniz-Institut für Evolutions- und Biodiversitätsforschung, Berlin 10115, Germany
[3]Natural Science Collection, Martin-Luther-University Halle-Wittenberg, Domplatz 4, Halle (Saale) D-06108, Germany
[4]Bulgarian Academy of Sciences, Institute of Biodiversity and Ecosystem Research, 1 Tsar Osvoboditel, Sofia 1000, Bulgaria
[5]Biologische Gutachten Dietz, Balinger Str. 15, 72401 Haigerloch, Germany
[6]Berlin-Brandenburg Institute of Advanced Biodiversity Research (BBIB), Altensteinstraße 6, 14195 Berlin, Germany

EÇ, 0000-0001-8188-8651

Secondary contacts can play a major role in the evolutionary histories of species. Various taxa diverge in allopatry and later on come into secondary contact during range expansions. When they meet, their interactions and the extent of gene flow depend on the level of their ecological differentiation and the strength of their reproductive isolation. In this study, we present the multilocus phylogeography of two cryptic whiskered bat species, *Myotis mystacinus* and *M. davidii*, with a particular focus on their putative sympatric zone. Our findings suggest that *M. mystacinus* and *M. davidii* evolved in allopatry and came into secondary contact during range expansions. Individuals in the area of secondary contact, in Anatolia and the Balkans, have discordant population assignments based on the mitochondrial and the nuclear datasets. These observed patterns suggest that the local *M. mystacinus* populations hybridized with expanding *M. davidii* populations, which resulted in mitochondrial introgression from the former. In the introgression area, *M. mystacinus* individuals with concordant nuclear and mitochondrial genotypes were identified in relatively few locations, suggesting that the indigenous populations might have been largely replaced by invading *M. davidii*. Changing

environmental conditions coupled with ecological competition is the likely reason for this replacement. Our study presents one possible example of a historical population replacement that was captured in phylogeographic patterns.

## 1. Introduction

Geographic distribution of species is complex expression of their ecology and evolutionary histories. Various factors, such as environmental conditions (abiotic), interaction with other taxa (biotic) and the ability to access areas that have the right set of abiotic and biotic factors, regulate the spatial distribution of species [1]. There is a widespread view that environmental components and biogeographical processes are the major factors that regulate species distributions at large spatial scales, whereas species interactions seem to have less importance [2]. However, the role of biotic factors in shaping biogeographic patterns remains largely unexplored [3].

Phylogeographic patterns of related species, especially in their contact zones, can provide insights about their interactions and how these interactions shape their evolutionary histories. Species usually diverge in allopatry and later on they can come into contact during range expansions. When they meet, their interactions depend on the combination of their ecological divergence and also the extent of their reproductive isolation. For instance, if taxa ecologically differentiated during their time in isolation, then they would be able to extend their ranges with minimal competition, eventually forming overlapping distributions. By contrast, if their niches still overlap, they would compete for the available resources; either forming allopatric distributions or one replacing the other. Pigot & Tobias [2] showed that in a diverse clade of passerine birds (Furnariida), rates of secondary sympatry were positively associated with both phylogenetic and morphological distance between species. As the time since speciation increases, the range expansions of sister species are more likely to lead to sympatry. This suggests that ecological competition is one of the major factors that shape the distribution patterns of related species.

Species interactions during range expansions also provide chances for genetic exchange. If taxa have not become fully reproductively isolated during their time in allopatry, then they might interbreed in their putative contact zones. Such gene flow might lead to introgression of genomic regions. Various studies show that genomic introgression occurred in diverse taxonomic groups [4–8], probably happening when species were colonizing new areas [9,10]. These introgression patterns suggest that species are not as reproductively isolated as once thought but may occasionally admix when they get into contact.

In this study, we investigate the evolutionary history of two cryptic whiskered bat species, *Myotis mystacinus* and *M. davidii*, with a particular focus on their putative sympatric zone. Whiskered bats, *M. mystacinus sensu lato* or referred as the *M. mystacinus* morphogroup, represent one of the most complex species groups of *Myotis* [11,12]. They include some of the smallest representatives of the genus *Myotis* and have a relatively conserved morphotype. Despite their morphological resemblance, these species are not monophyletic, but are scattered all around the species tree [13,14]. In the western Palaearctic, this morphogroup is represented by five species: (i) *M. mystacinus*; (ii) *M. davidii* (previously as *M. aurascens*; see [15]); (iii) *M. brandtii*; (iv) *M. alcathoe*, and (v) *M. hyrcanicus*. Genetically, *M. mystacinus* and *M. davidii* are closely related, as are *M. alcathoe* and *M. hyrcanicus*. These two pairs of sister species are deeply diverged from each other [12,13,16]. *Myotis brandtii* is even more distantly related and clusters within the New World clade of the genus *Myotis* [13]. In East Asia, there are other related species, such as *M. ikonnikovi* and *M. altarium* [13], yet relatively little is known about them.

Both of our study species have wide and largely allopatric distribution ranges: *M. mystacinus* spreads from northwest Africa in the west to the Caucasus in the east, and *M. davidii* ranges from the Balkans in the west to South Korea in the east. They can be separated from the other members of the morphogroup by their size and some dental characters [12,17]. However, their morphological distinction is rather difficult, especially in the Balkans and Anatolia, where both taxa are supposed to coexist [12,17]. In these regions, their distribution is poorly known [12,17–19]. Surprisingly, in this area, almost all the genetically screened individuals belong to the *M. mystacinus* mtDNA lineage [14,20], yet their nuclear DNA (nuDNA) have not been analysed.

Here, by using nuclear and mtDNA markers, we investigate the phylogeography of these species. We aim to (i) infer the evolutionary histories of *M. mystacinus* and *M. davidii*, (ii) investigate the discordance between the morphology and the mtDNA of the Balkan and the Anatolian populations, and (iii) resolve

the taxonomic identity of the populations in the putative sympatric range, especially the Bulgarian populations which were suggested as a distinct taxon, *M. mystacinus bulgaricus*.

# 2. Material and methods

## 2.1. Genetic data sampling

We generated genetic data for 292 samples collected from 125 localities (electronic supplementary material, appendix). DNA isolation and the sequencing of a partial mitochondrial gene, NADH dehydrogenase subunit 1 (ND1), were conducted as described in [21]. For mtDNA analysis, we generated 240 sequences (450 bp long) and analysed them with 133 sequences gathered from GenBank (electronic supplementary material, appendix). For nuDNA analysis, a subset of samples was sequenced for four nuclear introns—the intron 5 of ABHD11 gene ($n = 103$), the intron 3 of ACOX2 gene ($n = 156$), the intron 4 of COPS gene ($n = 136$) and the intron 7 of ROGDI gene ($n = 152$)—following [22] and [10], who previously used these markers for *M. nattereri* species complex (table 2).

Generated sequences were edited and aligned in CodonCode Aligner (CodonCode Corporation, MA, USA). We used the PHASE algorithm as implemented in DnaSP 5.10 [23] to phase the nuclear sequences. Estimates of genetic divergence among lineages were conducted in MEGA7 [24] and sequence polymorphism estimates in DnaSP. We used the HaploNet function as implemented in the R package Rpegas 0.10 [25] to generate haplotype networks. All the genetic data generated are deposited in GenBank with accession numbers MN706627–MN707417 (electronic supplementary material, appendix).

## 2.2. Phylogenetic reconstructions

For the mtDNA analysis, we reconstructed a Bayesian phylogenetic tree using BEAST 1.8.3 [26]. We ran two independent 20 million chains, both with coalescent exponential population and coalescent constant population priors, and HKY was selected as the substitution model. Convergence of the runs and the effective sample size (ESS) of the estimated parameters were checked in TRACER 1.6.0 (http://tree.bio.ed.ac.uk/software/tracer). In all the runs, all the parameters converged and they had ESS values higher than 200. For both population priors, results were similar; here we present the results of the coalescent exponential population prior. The resulting tree and log files of repeated runs were combined in LOGCOMBINER 1.8.0 (http://beast.bio.ed.ac.uk/logcombiner) with a 10% burn-in. The trees were summarized using the maximum clade credibility topology in TreeAnnotator and the final tree was rooted with *M. alcathoe*.

For the nuclear markers, we used two different approaches. In the first one, we used *BEAST [27] as implemented in BEAST 1.8.3 [26]. We assembled the samples in five groups, based on their mtDNA and also clustering in nuDNA (see Results §3.3): The Europe—including the Anatolian *mystacinus*—(1) and the Caucasus (2) *mystacinus* clades, the western (3) and the eastern (4) *davidii* clades and the genetically discordant individuals (5), which have *M. mystacinus* mtDNA but cluster with *M. davidii* in nuDNA. We randomly selected ten samples for each of these groups and used Yule model for the tree prior and HKY as the substitution model. Three independent runs of 50 million chains were computed and post-processed similar to the mtDNA BEAST runs.

In the second approach, we used the Bayesian Phylogenetics and Phylogeography program (BPP) v. 4 in the unguided species delimitation mode (A11) as described in [28]. The same dataset was used as in the *BEAST analysis. We used the 'algorithm 1' for the rjMCMC and selected 'uniform rooted trees' for the species model prior with using the default fine-tuning parameters. For the theta and the tau estimates, we assigned the inverse γ priors 'a' and 'b' as 3 and 0.02, respectively. We repeated these runs changing the inverse γ priors 'a' and 'b' for the tau estimates as 3 and 0.08. The runs were replicated two times and all of the estimates were consistent.

## 2.3. Population genetic structure

We used principal component analyses (PCA) and STRUCTURE 2.3.4 [29] to investigate the population structure. For these analyses, we used the allele data of phased sequences. For PCA, we used the R package adegenet 2.1.1 [30]. Samples which had data for at least three intron markers were used. For STRUCTURE analysis, we ran ten independent 100 000 chains and discarded the first 20% for $K = 1$–5. We used the admixture model with the correlated allele frequencies. We repeated the runs also

with the 'loc prior' model [31] assigning individuals into five populations, which were described in the *BEAST analysis section. These runs were merged in CLUMPAK server [32] and visualized in R.

In order to infer the level of population splits, we used two different approaches. In the first one, we calculated Evanno's ΔK [33] as implemented in CLUMPAK server, to estimate the optimal value of $K$ for the STRUCTURE runs. In the second approach, we calculated the genealogical divergence index ($gdi$) [34] as described in [35]. This index was proposed for delineating species and suggested that $gdi$ estimates less than 0.2 suggest a single species and $gdi$ estimates greater than 0.7 suggest distinct species, while $gdi$ values within the range indicate ambiguous delimitation [34]. For the $gdi$ estimates, we again used the BPP v. 4 with the previously described priors. But this time, we used A00 model, which estimates parameters on a fixed species phylogeny. Here, we used the phylogeny estimated in the previous A11 runs. As described in [35], sister populations inferred to belong to same species by $gdi$ are collapsed, and resulting species tree is used to conduct a new BPP analysis until root of the tree is reached.

## 2.4. Demographic analysis

We used Bayesian and extended Bayesian skyline analysis to estimate the demographic history of the lineages for mitochondrial and nuclear datasets, respectively. Both of these analyses were run in BEAST v. 2.5.1 [36] using the default priors and HKY as the substitution model. Runs were computed for 100 million chains. Their convergence and effective sample size (ESS) of the estimated parameters were checked in TRACER 1.7.1 (http://tree.bio.ed.ac.uk/software/tracer). In all the runs, all the parameters converged and they had ESS values higher than 200. We used TRACER 1.7.1 for the Bayesian skyline, and EBSPAnalyser for the extended Bayesian skyline reconstructions with a 10% burn-in.

# 3. Results

## 3.1. Mitochondrial phylogeography

Phylogenetic reconstruction of the ND1 sequences revealed a deeply divergent tree (figure 1c). We identified three lineages, which differ by approximately 11% in p-distances from each other. One lineage is distributed in the west, ranging from Northern Africa to the Caucasus (figure 1a). We refer to this lineage as the Clade M, which includes the M. mystacinus individuals from western Europe; in this area, M. mystacinus is allopatric from its sister species, M. davidii. Within the Clade M, some of the individuals from the Caucasus formed a subclade, which differed from the rest by approximately 3%. Within the former subclade, the largest distance is approximately 3% between the samples from Israel and Greece; within the Caucasus subclade, diversity is relatively lower, with approximately 1.4% highest divergence.

The second lineage is formed by M. ikonnikovi samples from Mongolia and Japan. They cluster with a third lineage, which covers a wide area, ranging from Bulgaria to South Korea. We refer to this latter lineage as the Clade D, which includes the M. davidii individuals from Mongolia (following [15]). This lineage is also composed of two subclades, which are geographically structured: the western Clade D is found to the north of the Black Sea, the Caucasus, northern Iran and Kyrgyzstan; the eastern Clade D is found in Mongolia and its bordering region from Russia, and South Korea (figure 1b). These subclades differ from each other approximately by 4%. Within the western Clade D, the highest divergence is around 2.2%, and within the eastern Clade D, it is 1.8%.

The distribution of the Clade M haplotypes shows a weak spatial structuring (figure 2). In the haplotype network, the European samples exhibit a star-like structure, where the majority of the haplotypes were connected to one of the central ones. These star-like structures are signatures for population expansion. Some haplotypes are widely distributed. For instance, the most common haplotype—haplotype 3—is found across a wide area, ranging from western Anatolia to southern Germany, again suggesting a recent population expansion (figure 2c). Samples from the most western ranges—from Ireland, Spain and Morocco—share the haplotypes that are found in Central Europe or that are closely related to them.

In the eastern ranges, in the Caucasus, Anatolia and the Balkans, populations have high haplotype and nucleotide diversities (table 1), and in the network, haplotypes are connected with more mutational steps. These patterns are signatures for older and more persistent populations. Samples from the western ranges, on the other hand, have lower nucleotide diversities and were connected to each other with few mutational steps suggesting a younger age.

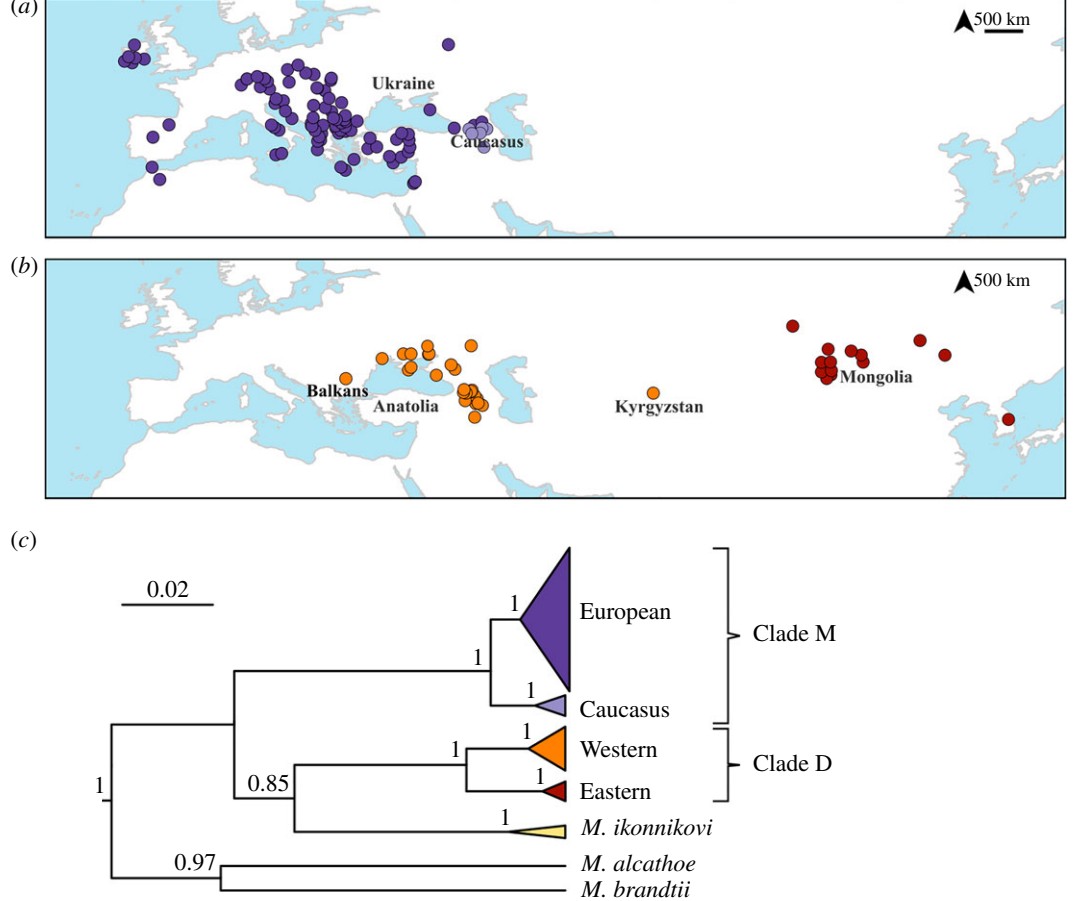

**Figure 1.** Mitochondrial phylogeography of the *M. mystacinus* and *M. davidii* complex. (*a*) Distribution of the Clade M and (*b*) distribution of the Clade D. (*c*) Bayesian phylogenetic tree of the partial ND1 gene.

Within the western Clade D, samples from the Caucasus have the highest nucleotide diversity (table 1). Samples from Kyrgyzstan belong to two haplotypes and are connected to the most common haplotype in the Caucasus (figure 3). Similarly, samples from the west and north of the Black Sea—Bulgaria, Ukraine and Russia—are very similar to a haplotype from the Caucasus.

## 3.2. Nuclear intron diversities

Among the four sequenced introns, the intron 3 of ACOX2 has the highest haplotype and nucleotide diversities (figure 4; table 2). Other introns have relatively lower and similar diversity values. The haplotype network of the intron 5 of ABHD11 has a star-like structure, in which almost all the haplotypes are connected to one central one. Other markers show more diverse patterns, yet none of them have a clear sorting for the mitochondrial lineages nor do they form two distinct clades (figure 5). Various haplotypes are shared by both of the mitochondrial lineages.

## 3.3. Population structure

We used STRUCTURE and PCA analyses to investigate population structuring. Both approaches revealed similar results. In the STRUCTURE analysis, both with and without loc prior models, samples group into two clear clusters for $K = 2$ (figure 5*b*; electronic supplementary material, figure S1). The first group is exclusively composed of the individuals which belong to the Clade M in mtDNA. This cluster is distributed in the Caucasus and Europe, but excluding most of the Balkans (figure 5*a*). One individual from Anatolia also clusters within this group. We did not identify any admixed individuals for $K = 2$. This might represent a lack of contemporary hybridization between these groups, but also might be affected by the resolution of analysed nuclear markers. Further studies, especially using genomic approaches can provide more definitive results.

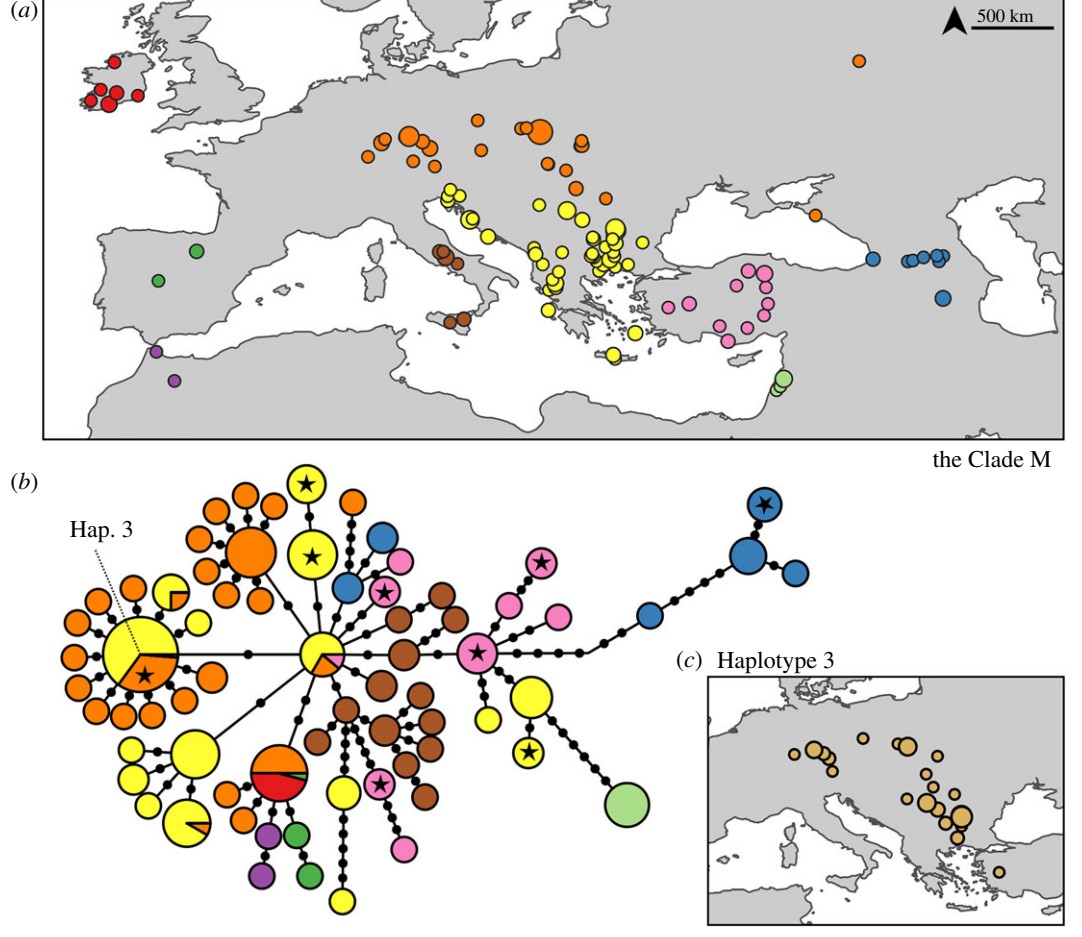

**Figure 2.** Phylogeography of the Clade M. (*a*) Distribution samples coloured based on their geographical locations. (*b*) Haplotype network of the Clade M. Haplotypes marked with stars are shared with *M. davidii*. (*c*) Distribution of the most common haplotype, Hap. 3. The sizes of the points and haplotypes are proportional to the sample sizes.

**Table 1.** Summary statistics for the partial ND1 gene: number of individuals (*N*), number of identified haplotypes (*h*) and segregating sites (ss), haplotype (*Hd*) and nucleotide (π) diversity estimates for each population.

| lineage | population | N | h | ss | Hd | π |
|---|---|---|---|---|---|---|
| Clade M | Ireland | 11 | 1 | 0 | 0.0000 | 0.0000 |
| | Morocco | 2 | 2 | 1 | 1.0000 | 0.0022 |
| | Spain | 3 | 3 | 2 | 1.0000 | 0.0030 |
| | C Europe | 78 | 26 | 30 | 0.8342 | 0.0050 |
| | Italy | 16 | 13 | 12 | 0.9750 | 0.0067 |
| | Balkans | 121 | 16 | 25 | 0.7930 | 0.0056 |
| | Turkey | 18 | 10 | 17 | 0.8824 | 0.0088 |
| | Israel | 9 | 1 | 0 | 0.0000 | 0.0000 |
| | Caucasus | 13 | 6 | 17 | 0.8590 | 0.0165 |
| Clade D | N Black Sea | 15 | 5 | 3 | 0.4762 | 0.0012 |
| | Caucasus | 38 | 7 | 14 | 0.5832 | 0.0058 |
| | Kyrgyzstan | 4 | 2 | 2 | 0.5000 | 0.0022 |
| | Mongolia | 24 | 8 | 14 | 0.6486 | 0.0042 |

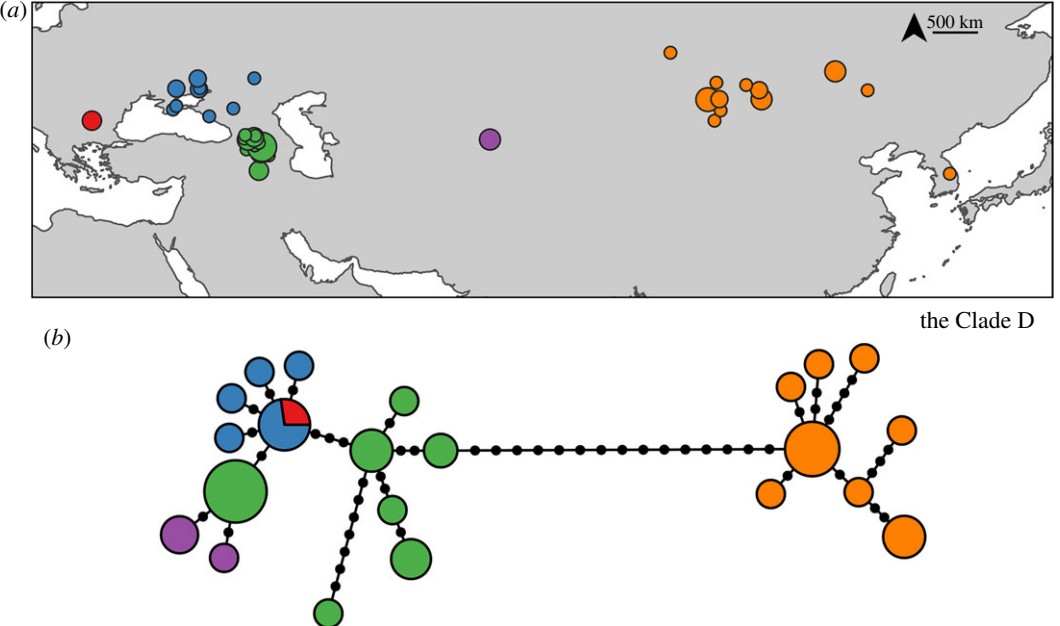

**Figure 3.** Phylogeography of the Clade D. (*a*) Distribution samples coloured based on their geographical locations. (*b*) Haplotype network of the Clade D. The sizes of the points and haplotypes are proportional to the sample sizes.

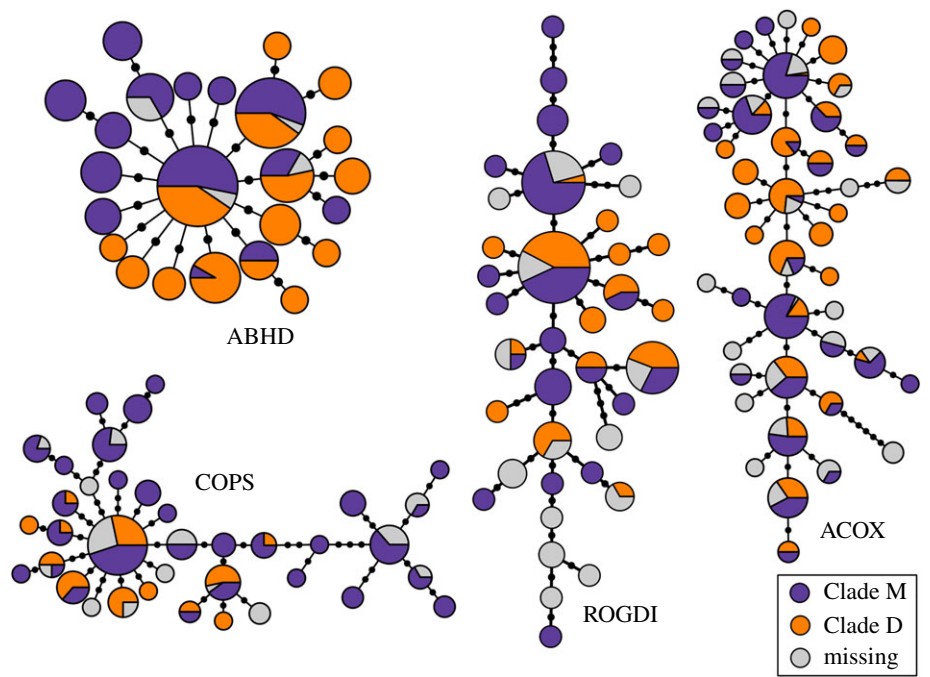

**Figure 4.** Haplotype networks of the analysed nuclear introns. The sizes of the haplotypes are proportional to the sample sizes and their colours are based on the mtDNA assignment. Individuals which did not have mtDNA sequences are coloured grey.

**Table 2.** Polymorphism information about the analysed nuclear introns; length after removing gaps (bp), number of individuals (*N*), number of variable sites (*S*), number of haplotypes (*h*), haplotype diversity (*Hd*) and nucleotide diversity (π).

| gene | length (bp) | N | S | h | Hd | π |
|---|---|---|---|---|---|---|
| ABHD11 intron 5 | 330 | 103 | 19 | 24 | 0.7948 | 0.0038 |
| ACOX2 intron 3 | 142 | 156 | 23 | 45 | 0.9307 | 0.0206 |
| COPS intron 4 | 621 | 136 | 33 | 37 | 0.7666 | 0.0048 |
| ROGDI intron 7 | 307 | 152 | 27 | 37 | 0.7638 | 0.0062 |

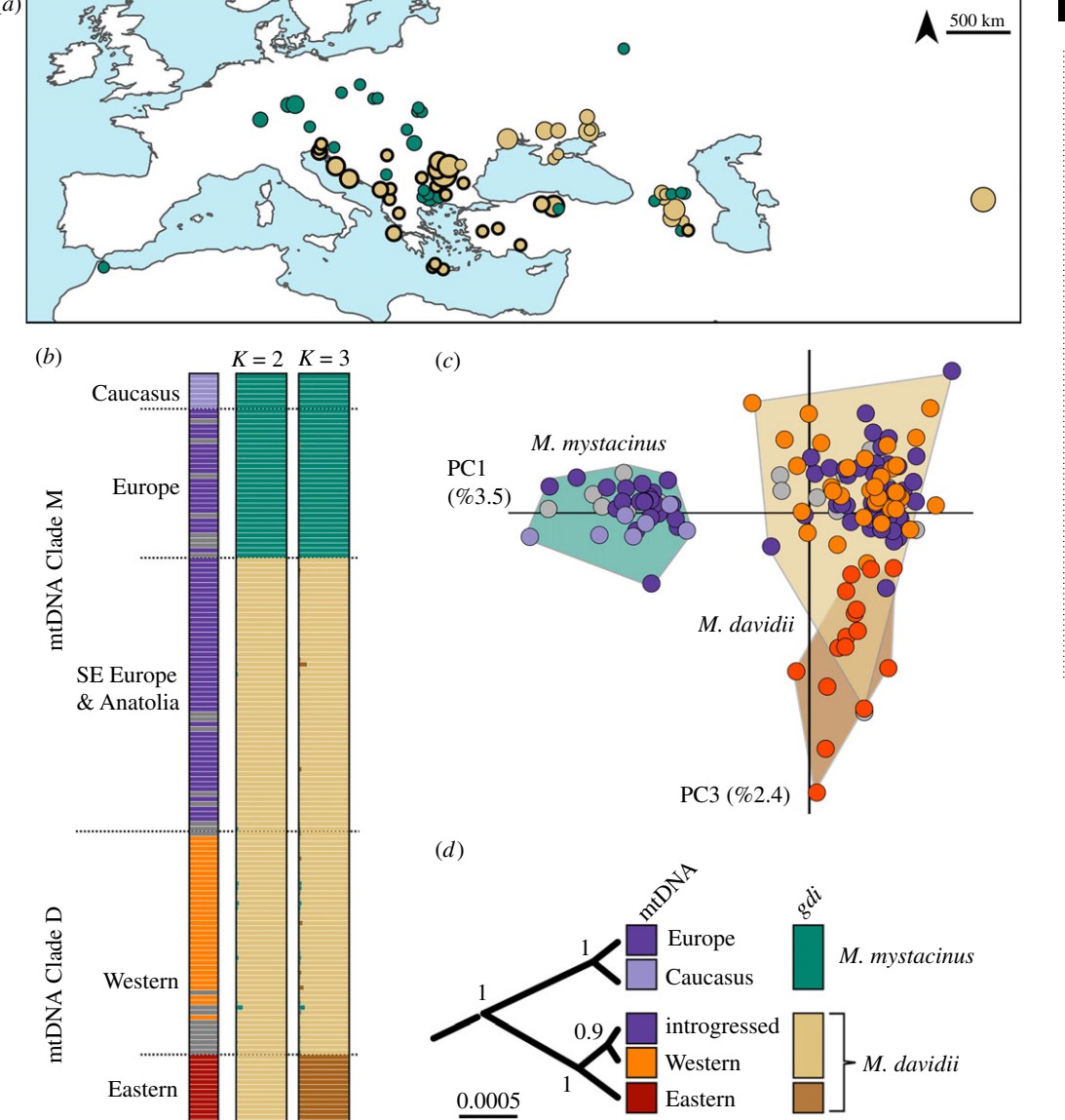

**Figure 5.** Population structure based on the nuclear introns. (*a*) Distribution of the identified nuclear clades, excluding the eastern Clade D. Circles which have bold outlines indicate discordant assignment based on mtDNA and nuDNA datasets. The sizes of the points are proportional to the sample sizes. (*b*) STRUCTURE results for *K* = 2 and *K* = 3 for the loc prior model. The left panel shows the mtDNA assignments of the analysed samples. (*c*) PCA; circles are coloured based on the mtDNA assignments and shaded polygons are based on the STRUCTURE run for *K* = 3. In (*b*) and (*c*), individuals which did not have mtDNA sequences are coloured grey. (*d*) *Beast species tree based on nuclear introns, also showing the species delineation based on the *gdi* estimations.

The second cluster is composed of all the individuals that belong to the Clade D in mtDNA, but also includes individuals which belong to the Clade M. This cluster has a wide distribution, ranging from Slovenia in the west to Mongolia in the east. Samples which have discordant mtDNA are found in the Balkans, Anatolia and the Caucasus (figure 5*a*). For *K* = 3, the second group splits further into two, where the eastern Clade D individuals form a distinct cluster. In the runs without the loc prior, this group also included some samples from the west and structuring was not as discrete as in the loc prior model (figure 5*b*; electronic supplementary material, figure S1). For both of the models, with or without loc prior, the optimal *K* value estimated by Evanno method was identified as two.

The PCA analysis revealed a similar structuring (figure 5*c*). The first principal component splits *M. mystacinus* and *M. davidii* (variance explained 3.5%), and the third component mostly splits the eastern Clade D from the western Clade D (variance explained 2.4%); similar to the STRUCTURE result for *K* = 3. The second component explains the variation within the western Clade D (variance explained 2.5%) (electronic supplementary material, figure S2). We ran another PCA for *M. davidii*

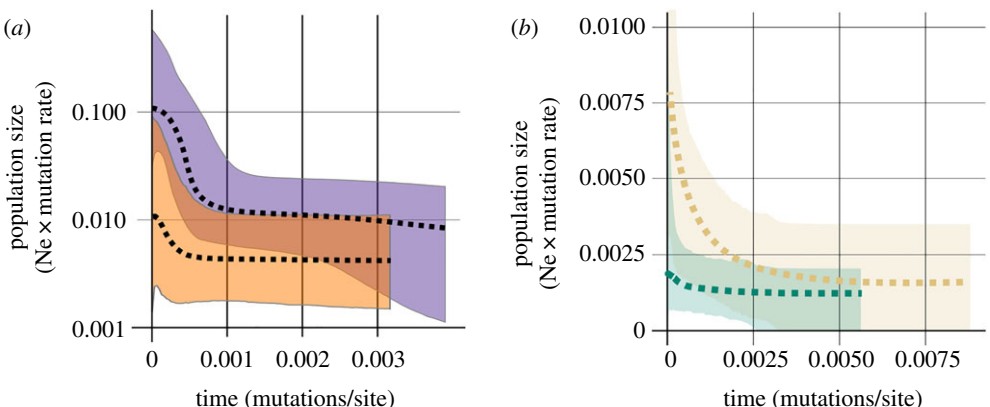

**Figure 6.** Population expansion time estimates. (*a*) Bayesian skyline plot for the European Clade M (purple) and the western Clade D (orange) based on the mtDNA sequences. (*b*) Extended Bayesian skyline plots for *M. mystacinus* in Europe (green) and the western Clade D (yellow), based on the nuDNA sequences. Dashed lines show mean estimates and shaded areas 95% highest posterior density of the estimates. The unit of *x*-axis is in mutations/site, and *y*-axis is in population size (Ne) × mutation rate. Note that both graphs have different ranges and the *y*-axis is in log scale in (*a*).

samples only (electronic supplementary material, figure S3). This analysis showed that the Caucasus and Mongolia harboured high diversity, and the rest of the regions, including the area of discordance, clustered as a subset within the Caucasus samples. Overall, 48 individuals had discordant assignments *M. davidii* nuclear background with clade M mtDNA, and 78 were concordant.

## 3.4. Nuclear phylogeny

Both the *BEAST and the A11 runs of BPP revealed the same phylogenetic reconstruction for nuclear markers. There were two major clusters: (i) the Europe and the Caucasus lineages of the Clade M group together, and (ii) the Clade D lineages and the majority of the populations from Anatolia and the Balkans which belonged to the Clade M clustered together. In the second cluster, the eastern Clade D is further split from the rest (figure 5*d*).

The A11 analysis identified all of these clades as distinct groups. However, the *gdi* estimates group them into three clades, suggesting that they might represent three distinct species: (i) the Europe and the Caucasus lineages of the Clade M were collapsed, both had *gdi* less than 0.1; (ii) the western Clade D and the Anatolian and the Balkan populations with the introgressed *M. mystacinus* mtDNA were collapsed together, both had low *gdi* estimates less than 0.1; and (iii) the eastern Clade D, which had *gdi* > 0.9 (figure 5*d*). We have a sampling gap from the eastern ranges, between the possible contact area of the western and the eastern Clade D lineages. The deep separation signal identified by *gdi* might be related to this and further studies are needed to investigate the taxonomic status of the Clade D subclades.

## 3.5. Demographic analysis

For the mtDNA sequences, Bayesian skyline analysis for the Clade M—excluding its Caucasian subclade—shows a recent population expansion. For a 4.7% divergence rate per million years [37], this expansion dates back to around 20 000 years ago (approximately starting at 0.0005 mutations/site ago in figure 6). There is a similar expansion signal for the western Clade D.

For the extended Bayesian skyline analysis, we used nuclear markers. We divided the data into two sets based on the nuclear clusters: *M. mystacinus* in Europe and the western Clade D populations. Here, the expansion signature of *M. mystacinus* was relatively weak compared to mtDNA (figure 6). *Myotis davidii* showed a clearer expansion pattern; both starting at relatively the same time.

## 4. Discussion

We identified genetically distinct clusters both in mtDNA and nuDNA. Assignment of some of the individuals differed based on the dataset. Individuals with such discordant assignments were identified in the Balkans, Anatolia and the Caucasus. In these regions, almost all the analysed individuals have the mtDNA of the clade M, yet in nuDNA, they cluster with the *M. davidii* samples.

Such discordant patterns can be explained either with incomplete lineage sorting or introgression. Considering that the discordant individuals are located in the contact zone of the *M. mystacinus* and the *M. davidii* mtDNA lineages, it is likely that this spatial pattern resulted from introgression events rather than incomplete lineage sorting. In the latter, discordant individuals would be expected to be randomly distributed rather than being geographically structured [38]. We have two possible introgression scenarios that can explain the observed pattern of discordance.

## 4.1. Breakdown of reproductive isolation

In the first scenario, we presume that the past distribution ranges of both species were similar to their current ones, with an overlap ranging from the Balkans to the Caucasus. At some point, the reproductive barrier between the sympatric populations broke and they started to hybridize. This might have happened during a period of environmental change, when one species became advantageous and started to increase its population size, whereas the other one possibly started to decline. In such a scenario, there would be more gene flow from the expanding taxon to the other, shifting the nuDNA profile of the admixed populations to resemble the former. If there are no sex specific mating preferences, their mtDNA, on the other hand, would be inherited from both of the taxa. However, we found that all the discordant individuals from the Balkans and Anatolia have the mtDNA of *M. mystacinus*. Additionally, the introgressed mtDNA shows geographical structuring, suggesting that it happened independently at multiple locations. This pattern would be expected if the mtDNA of *M. mystacinus* had a selective advantage or genetic drift would have acted in the same directions for all the admixed populations. Both the exclusive presence of mtDNA of *M. mystacinus* among discordant individuals and the evidence for multiple introgression events suggest that the scenario of reproductive isolation breakdown is unlikely.

## 4.2. Mitochondrial introgression during range expansions

In the second scenario, we presume that the ancestral origin of *M. davidii* is located in the east, probably further east than the Caucasus. At some point, this ancestral taxon expanded its range towards the west. First contact of these expanding *M. davidii* and the local *M. mystacinus* populations was probably in the Caucasus. Here, we found that both species occur and their gene pools remained unmixed. Later on, *M. davidii* expanded its range further to the west, possibly through Anatolia. These expanding populations admixed with local *M. mystacinus* populations and acquired at least their mtDNA. Later, they moved further westwards, and this time, they acquired the mtDNA from the Balkan populations. This westward expansion pattern is supported by the nuclear structuring of the *M. davidii* populations. Thus, our findings match the scenario of a westward range expansion of *M. davidii* accompanied with multiple mitochondrial introgression of *M. mystacinus*.

Admixture events during range expansions are one of the major mechanisms that can lead to observed introgression patterns [9,39]. Similar cytonuclear discordances are observed among other *Myotis* species [40,41], some also occurring in the same geographical areas such as the Balkans, Anatolia and the Caucasus [10,42]. Such exchange of genes might have fuelled species radiations by contributing to their adaptations in changing environments [43].

## 4.3. Expansion of *Myotis mystacinus*

One interesting question is the timing of the *M. mystacinus* expansion: did it happen before *M. davidii* invaded the Balkans or after? If the latter is the case, could adaptive introgression have played a role in its rapid expansion?

Demographic analysis showed that the extant *M. mystacinus* populations in Europe originated relatively recently from the Balkans. Bayesian skyline analysis of mtDNA sequences estimate a population size increase within the last 25 000 years, possibly starting after the Last Glacial Maximum. The western *M. davidii* lineage shows a similar expansion pattern, yet less pronounced. Here, we did not include the expansion time of *M. davidii* populations in Anatolia and the Balkans using mtDNA, since in these regions, their mtDNA was introgressed from *M. mystacinus*, along with their signatures of population demography.

To compare the relative timing of the *M. mystacinus* and the *M. davidii* expansions, we used nuDNA sequences. In the extended Bayesian skyline analysis, the signal of *M. mystacinus* expansion was weaker than the signal in mtDNA. This might be because of the relatively low diversity of nuclear markers and

the numbers of markers that we used. The *M. davidii* populations, on the other hand, showed a stronger signal for population expansion. The timing of these expansions is relatively the same.

The similarity of the expansion time estimates suggests that the range expansion of *M. mystacinus* started before the arrival of *M. davidii* in the Balkans. Accordingly, we do not consider that a genetic exchange occurred between them, prior to the expansion of *M. mystacinus*. Yet, our analysis is based on very few markers and these expansion time estimates require further investigation, preferably with genome wide approaches.

## 4.4. Taxonomic identity of the Balkan populations: Is *M. m. bulgaricus* a valid taxon?

The taxonomic identity of the Balkan populations has been an issue of debate for a long time (addressed in [12,14,19]). The uncertainty about their classification was caused by extensive morphological variation in the region. Only in 2001, *M. alcathoe* was described as a new species [44] and explained some of the morphological variation. As individuals with *M. mystacinus* mtDNA were still morphologically heterogeneous, some representing the nominotypical form and some *M. davidii*, the latter were considered to represent a separate taxon, *M. m. bulgaricus* [14,17,45]. Our analysis suggests that the presence of these different mtDNA lineages resulted from complex biogeography of this species group, including mitochondrial introgression and different colonization routes. Accordingly, we conclude that *M. m. bulgaricus* is a synonym of *M. davidii*. In the case that further studies would identify that the subclades of Clade D are distinctive at a taxonomic level, then the name *aurascens* should be recalled to name the western Clade D.

## 4.5. Population replacement in Anatolia and the Balkans

In the area of introgression, in few locations, we identified individuals that are assigned as *M. mystacinus* in nuclear markers. This finding is consistent with previous studies, which used morphological characters to discriminate these species. *Myotis davidii* is the most widespread and also the most abundant form in the Caucasus and adjacent areas [12], in Anatolia [18] and in the Balkans [11]. This pattern, *M. davidii* being the more abundant taxon in the introgression zone, raises a question about the historical *M. mystacinus* populations in this wide area: what happened to them?

Changing environmental conditions might have initiated the local extinction of *M. mystacinus* by forcing them to compete with the expanding *M. davidii*. Previous studies show that both species have relatively high plasticity in their ecological preferences (it has to be noted that these studies might have confused the species identifications, especially in the introgression area). In the Caucasus, *M. davidii* occupies various ecosystems, from arid lowland steppes to mountain steppes and forest, although with a certain preference for open habitats [12]. In central Europe, *M. mystacinus* is found in open and semi-open landscapes with isolated patches of woodland and hedgerows [17]. In Ireland, it was found to prefer mixed woodland and riparian habitats [46], in contrast with another study from Britain where only pasture was used [47]. In the Balkans, *M. mystacinus* is mostly restricted to the higher elevations in the mountainous regions [11]. It is likely that *M. davidii* has a competitive advantage in the Mediterranean habitats, which are characterized by a warmer and more arid climate, and hence replace the *M. mystacinus* populations in such habitats. Further ecological studies focusing to these contact zones can elucidate their niche preferences.

Our study demonstrates how species interactions, both in terms of gene flow and interspecific competition, play a role in the evolutionary history of related species and how these interactions can be inferred from phylogeographical patterns. Various studies identified evidence for past gene flow in diverse taxonomic groups including birds [4], cats [6], fishes [8] and humans [5]. Such genetic introgressions suggest that species interactions were widespread and probably had a crucial role in shaping the evolutionary history of species. Using a multilocus phylogeographic approach can elucidate the role of species interactions in shaping the current distribution patterns of related taxa.

Ethics. We used the DNA samples from the collection of Museum für Naturkunde. Additionally, wing punches from Bulgaria were collected by Heliana Dundarova with permits issued by the Bulgarian Ministry of Environment and Water (nos. 452/20.04.2012, 525/03.06.2013 and 575/10.04.2014) from Israel by Eran Amichai with the permissions of Israel Nature and National Parks Protection Authority (39607/2013).

Data accessibility. All the genetic data generated are deposited in GenBank with accession numbers MN706627–MN707417.

Authors' contributions. E.C. designed the study, carried out the analyses and drafted the manuscript. H.D. participated in the design of the study, collected field data and participated in the data analyses. C.D. participated in the design of the

study and collected field data. F.M. conceived of the study, designed the study, coordinated the study and helped draft the manuscript. All authors gave final approval for publication and agree to be held accountable for the work performed therein.

Competing interests. We declare we have no competing interests.

Funding. No funding has been received for this article.

Acknowledgements. In memory of our colleague and friend, the late Boyan Petrov. We are grateful to Isabelle Waurick and Sofia Hayden; they undertook the DNA extractions and performed sequencing. We would like to thank Ivana Budinski, Stanimira Deleva, Suren Gazaryan, Panagiotis Georgiakakis, Astghik Ghazaryan, Lena Godlevska, Stoyan Goranov, Otto von Helversen, Richard Hoffmann, Maria Jerabek, Marina Kipson, Kseniia Kravchenko, Friedo Kretzschmar, Radek Lučan, Elena Papadatou, George Papov, Primoz Presetnik, Guido Reiter, Bernd-Ulrich Rudolph, Konrad Sachanowicz, Wolfram Schulze, Niya Toshkova, Anton Vlaschenko, Maja Zagmajster, Violeta Zhelyazkova and various other researchers for their help during the fieldwork and/or getting access to tissue samples. We would like to thank Mozes Blom and Darija Josić for their comments on the manuscript, and Ulla Lächele and Anne Hänel for discussions about morphological comparisons. We are also grateful to Bridgit Schofield and Henry Schofield for language editing and proofreading.

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
