## [Reviewer comments · Royal Society Open Science]

Review History

RSOS-191805.R0 (Original submission)

Review form: Reviewer 1

Is the manuscript scientifically sound in its present form?

Yes

Are the interpretations and conclusions justified by the results?

Yes

Is the language acceptable?

Yes

Do you have any ethical concerns with this paper?

No

Have you any concerns about statistical analyses in this paper?

No

Recommendation?

Major revision is needed (please make suggestions in comments)

Comments to the Author(s)

Comments:

Although MtDNA introgression between lineages in bats is now very common, this MS is still very interesting considering population replacement between two bat species. The logic of this MS is good. But I have some major concerns before this MS is published.

1) Numerous studies have now shown the possible occurrence of selection on mtDNA. So I wonder whether the authors have conducted positive selection analyses on ND1 which was used in their phylogeographic study? If ND1 here is not neutral, it should not be used for phylogeographic or demographic analyses.

2) I have a little concern about ncDNA results. In the network analysis, there is no clear patterns, possibly due to the small number of polymorphic sites in these four introns. However, in the population structure and nuclear phylogeny, there are clear structuring or discordant clads. Because ncDNA results here are very important to assign admixed individuals in the contact regions of the two species, (if possible) I suggest the authors to sample more number of nuclear loci to confirm the current results.

3) In each map of this MS, there is no any information of localities. If not familiar to the geography of the studied regions, it is very difficult to follow the results and discussion. Suggest to add names for some important regions in this study.

4) In the discussion, there are too many speculations on evolutionary history of divergence and gene flow between the two studied species. I suggest the authors to conduct some formal demographic model tests (such as using Approximate Bayesian computation approach).

5) In the discussion, the authors mainly focus on their studied species and no general implications of this study (e.g. how species interactions can be inferred from their phylogeographic patterns and how their interactions affect their distributions) have been discussed in a wide range of animals or plants, which is important for other readers who do not study bats.

Minor comments:

1) 2.1. Genetic data sampling. Line 113: "a subset of samples", suggest to provide specific number here.

2) 3.5 Demographic analysis. What is the divergence rate for ncDNA used here?

Review form: Reviewer 2

Is the manuscript scientifically sound in its present form?

Yes

Are the interpretations and conclusions justified by the results?

Yes

Is the language acceptable?

No

Do you have any ethical concerns with this paper?

No

Have you any concerns about statistical analyses in this paper?

Yes

Recommendation?

Major revision is needed (please make suggestions in comments)

Comments to the Author(s)

The authors present results of a genetic survey based on wing punches collected on whiskered bats sampled across Eastern Europe, the Middle East and Central Asia. They used one mitochondrial (ND1) and four nuclear introns to test phylogeographic patterns and discovered that these bats are represented by two major, non-coincident lineages according to these markers. The discordant pattern between the two classes of markers is located in the Balkans and in Anatolia, while animals from elsewhere in Europe and the Caucasus show complete concordance. Although none of the tested nuclear marker shows any major subdivision of alleles (all four show more or less a star-like structure of networks), when analysed together, they support the existence of two discrete clusters, corresponding each to distinct species (*M. mystacinus* and *M. davidii*, respectively). The authors interpret the areas of discordance as resulting from past events of (asymmetric) introgression leading to a replacement of the mitochondrial genome of one species into the other species. Some rough estimates of expansion time are also given, but they are quite basic (no variance around molecular rates or in estimates is taken into consideration). In the discussion the authors try to fit these results to two narrative scenarios, one of which seems more likely (post LGM expansion of one species into the range of the other).

These results are interesting as they cover a significant part the Western Palearctic distribution of those two species and include both mtDNA and nucDNA markers in their analyses. However, given the complexity of the patterns of introgression in these two classes of markers, I struggled a lot to keep track of which analysis the author were refereeing to in the text, as they used “*mystacinus*” or “*davidii*” to designate either the species themselves (but how were they identified a priori is a mystery), to refer to mitochondrial clades, or to refer to nuclear “clades” (these are indeed not clades in a cladistic sense; clusters or groups should be used instead). The lack of clarity is blatant in the figure legends (e.g. meaning of the stars in Fig. 2b) and in the discussion, and does not help for a smooth understanding of their points. It is also unclear what they call “eastern *davidii*” as the sampling covered e.g. in Fig. 3 or in Fig. 5 obviously varies. The same is true for the meaning of “N” mentioned in the Table 1 (=individuals analysed?) and Table 2 (no idea what it means, individuals, alleles, etc.). Perhaps a good way to improve the clarity throughout the manuscript would be to simply name the mtDNA haplogroups “Clade 1” and “Clade 2”, restrain the names “*mystacinus*” and “*davidii*” to the two nuclear clusters, and use the term “alleles” for the nuclear variants and “haplotypes” for the mtDNA one (I know, this is sometimes also confusing in the existing literature). Finally, you could just equate these two clusters to the two species, avoiding the confusing ID issued from mtDNA vs nucDNA vs morphology.

Regarding the methods, it appears that STRUCTURE is of fundamental importance in clustering the multilocus genotypes into two groups. Hence all necessary parameters need to be detailed, which is not the case. How many K were tested? Did you include for instance K =1? How many replicates for each K (3 is by far insufficient; usually 10-20 are more meaningful)? It should also be mentioned and justified whether you used the admixture model and the correlated allele frequencies, or not, and whether you restricted your analyses to alleles of nuclear introns only. A good application of these methods in the context of gene introgression can be found in e.g. ESSELSTYN et al. 2013. Carving out turf in a biodiversity hotspot: multiple, previously unrecognized shrew species co-occur on Java Island, Indonesia. *Molecular Ecology* 22:4972-4987. I see that some important results (*BEAST) are not shown but should be given at least in the suppl. info..

Besides these general considerations, I have made a number of more minor suggestions/corrections/questions near the following line numbers:

36 ...indigenous?

60 ...extent rather than strength?

61 ...evolved towards different ecologies while in allopatry?
 75 separated or reproductively isolated?
 95 their morphological distinction is difficult; supposed to coexist
 98 avoid double yet
 114 the four nuclear introns must be named in full here, not only abbreviated in various tables and figures.
 136 How did you manage the phased alleles for the *BEAST and BPP analyses?
 139 What about the Anatolian mystacinus?
 183 divergent
 243 How many samples are discordant? Perhaps this important information should be highlighted in Fig. 5a with different symbols? That important part of the Fig. 5a is anyway impossible to understand, one cannot see the colour of the circles anyway!
 253 I find it strange that you did not use the second largest component, but the third for this figure. Can you justify this choice? Or does it appear in the suppl. mat.?
 255 Sorry I cannot understand this sentence
 263 ... lineage is further split ...
 264 There are now 5 clades??? 2 major + 1 = 3! Please rephrase this part to offer a clearer text for the reader.
 271 as the type locality for davidii is China, if anything, it should be the Western Pal davidii that would need additional taxonomic scrutiny (contra e.g. your ref. 15). And before suggesting to use the name bulgaricus, a number of older ones should be evaluated, e.g. in the Caucasus (see your ref. 12 and 15).
 280 It is unclear how you sorted the samples prior to analyses. Probably according to STRUCTURE clusters?
 286 How many concordant and how many discordant?
 308 and 312 if the introgression really happened several times independently, then this could also suggest a random lineage sorting, which would cancel your argument about that possibility earlier in the text.
 318 this seems to contradict your hypothesis of an old, stable pop (l. 213).
 326 along PCA3?
 Fig. 4 (l. 585) What is the meaning of the "missing" category? How did you treat these "missing" genotypes in the STRUCTURE and BPP analyses?
 367-369 given the continuity of nuclear genotypes between the Balkans and the Caucasus, there are more ancient names available to name this taxon, before suggesting the use of bulgaricus!
 371 "genetically" is too vague in this context. Did you mean nuclearly?
 380 High can you refer to previous studies on ecological plasticity of these two species, while you present here for the first time a new taxonomic arrangement? Anything presented so far should have confused the animals in the Balkans, giving a false idea of variation.

Decision letter (RSOS-191805.R0)

06-Feb-2020

Dear Dr Coraman,

The editors assigned to your paper ("Patterns of mtDNA introgression suggest population replacement in Palearctic whiskered bat species") have now received comments from reviewers. We would like you to revise your paper in accordance with the referee and Associate Editor suggestions which can be found below (not including confidential reports to the Editor). Please note this decision does not guarantee eventual acceptance.

Please submit a copy of your revised paper before 29-Feb-2020. Please note that the revision deadline will expire at 00.00am on this date. If we do not hear from you within this time then it

will be assumed that the paper has been withdrawn. In exceptional circumstances, extensions may be possible if agreed with the Editorial Office in advance. We do not allow multiple rounds of revision so we urge you to make every effort to fully address all of the comments at this stage. If deemed necessary by the Editors, your manuscript will be sent back to one or more of the original reviewers for assessment. If the original reviewers are not available, we may invite new reviewers.

- Data accessibility

If you wish to submit your supporting data or code to Dryad (<http://datadryad.org/>), or modify your current submission to dryad, please use the following link:
<http://datadryad.org/submit?journalID=RSOS&manu=RSOS-191805>

- Competing interests

- Authors' contributions

- Acknowledgements

- Funding statement

Kind regards,

Andrew Dunn

on behalf of Dr David Ferrier (Associate Editor) and Kevin Padian (Subject Editor)

Editor comments:

Thank you for your submission. We recognize that this is a complex problem with many dimensions, and the reviewers have identified some areas that require further clarity or explanation. Please address these specifically as you revise, and if you need more time contact our editorial office. Best wishes.

Comments to Author:

Reviewers' Comments to Author:

Reviewer: 1

Comments to the Author(s)

Comments:

Although MtDNA introgression between lineages in bats is now very common, this MS is still very interesting considering population replacement between two bat species. The logic of this MS is good. But I have some major concerns before this MS is published.

1) Numerous studies have now shown the possible occurrence of selection on mtDNA. So I wonder whether the authors have conducted positive selection analyses on ND1 which was used in their phylogeographic study? If ND1 here is not neutral, it should not be used for phylogeographic or demographic analyses.

2) I have a little concern about ncDNA results. In the network analysis, there is no clear patterns, possibly due to the small number of polymorphic sites in these four introns. However, in the population structure and nuclear phylogeny, there are clear structuring or discordant clads. Because ncDNA results here are very important to assign admixed individuals in the contact regions of the two species, (if possible) I suggest the authors to sample more number of nuclear loci to confirm the current results.

3) In each map of this MS, there is no any information of localities. If not familiar to the geography of the studied regions, it is very difficult to follow the results and discussion. Suggest to add names for some important regions in this study.

4) In the discussion, there are too many speculations on evolutionary history of divergence and gene flow between the two studied species. I suggest the authors to conduct some formal demographic model tests (such as using Approximate Bayesian computation approach).

5) In the discussion, the authors mainly focus on their studied species and no general implications of this study (e.g. how species interactions can be inferred from their phylogeographic patterns and how their interactions affect their distributions) have been discussed in a wide range of animals or plants, which is important for other readers who do not study bats.

Minor comments:

1) 2.1. Genetic data sampling. Line 113: "a subset of samples", suggest to provide specific number here.

2) 3.5 Demographic analysis. What is the divergence rate for ncDNA used here?

Reviewer: 2

Comments to the Author(s)

The authors present results of a genetic survey based on wing punches collected on whiskered bats sampled across Eastern Europe, the Middle East and Central Asia. They used one mitochondrial (ND1) and four nuclear introns to test phylogeographic patterns and discovered that these bats are represented by two major, non-coincident lineages according to these markers. The discordant pattern between the two classes of markers is located in the Balkans and in Anatolia, while animals from elsewhere in Europe and the Caucasus show complete concordance. Although none of the tested nuclear marker shows any major subdivision of alleles (all four show more or less a star-like structure of networks), when analysed together, they support the existence of two discrete clusters, corresponding each to distinct species (*M. mystacinus* and *M. davidii*, respectively). The authors interpret the areas of discordance as resulting from past events of (asymmetric) introgression leading to a replacement of the mitochondrial genome of one species into the other species. Some rough estimates of expansion time are also given, but they are quite basic (no variance around molecular rates or in estimates is taken into consideration). In the discussion the authors try to fit these results to two narrative scenarios, one of which seems more likely (post LGM expansion of one species into the range of the other).

These results are interesting as they cover a significant part the Western Palearctic distribution of those two species and include both mtDNA and nucDNA markers in their analyses. However, given the complexity of the patterns of introgression in these two classes of markers, I struggled a lot to keep track of which analysis the author were refereeing to in the text, as they used "mystacinus" or "davidii" to designate either the species themselves (but how were they identified a priori is a mystery), to refer to mitochondrial clades, or to refer to nuclear "clades" (these are indeed not clades in a cladistic sense; clusters or groups should be used instead). The lack of clarity is blatant in the figure legends (e.g. meaning of the stars in Fig. 2b) and in the discussion, and does not help for a smooth understanding of their points. It is also unclear what they call "eastern davidii" as the sampling covered e.g. in Fig. 3 or in Fig. 5 obviously varies. The same is true for the meaning of "N" mentioned in the Table 1 (=individuals analysed?) and Table 2 (no idea what it means, individuals, alleles, etc.). Perhaps a good way to improve the clarity throughout the manuscript would be to simply name the mtDNA haplogroups "Clade 1" and "Clade 2", restrain the names "mystacinus" and "davidii" to the two nuclear clusters, and use the term "alleles" for the nuclear variants and "haplotypes" for the mtDNA one (I know, this is sometimes also confusing in the existing literature). Finally, you could just equate these two

clusters to the two species, avoiding the confusing ID issued from mtDNA vs nucDNA vs morphology.

Regarding the methods, it appears that STRUCTURE is of fundamental importance in clustering the multilocus genotypes into two groups. Hence all necessary parameters need to be detailed, which is not the case. How many K were tested? Did you include for instance K =1? How many replicates for each K (3 is by far insufficient; usually 10-20 are more meaningful)? It should also be mentioned and justified whether you used the admixture model and the correlated allele frequencies, or not, and whether you restricted your analyses to alleles of nuclear introns only. A good application of these methods in the context of gene introgression can be found in e.g. ESSELSTYN et al. 2013. Carving out turf in a biodiversity hotspot: multiple, previously unrecognized shrew species co-occur on Java Island, Indonesia. *Molecular Ecology* 22:4972-4987. I see that some important results (*BEAST) are not shown but should be given at least in the suppl. info..

Besides these general considerations, I have made a number of more minor suggestions/corrections/questions near the following line numbers:

36 ...indigenous?

60 ...extent rather than strength?

61 ...evolved towards different ecologies while in allopatry?

75 separated or reproductively isolated?

95 their morphological distinction is difficult; supposed to coexist

98 avoid double yet

114 the four nuclear introns must be named in full here, not only abbreviated in various tables and figures.

136 How did you manage the phased alleles for the *BEAST and BPP analyses?

139 What about the Anatolian mystacinus?

183 divergent

243 How many samples are discordant? Perhaps this important information should be highlighted in Fig. 5a with different symbols? That important part of the Fig. 5a is anyway impossible to understand, one cannot see the colour of the circles anyway!

253 I find it strange that you did not use the second largest component, but the third for this figure. Can you justify this choice? Or does it appear in the suppl. mat.?

255 Sorry I cannot understand this sentence

263 ... lineage is further split ...

264 There are now 5 clades??? $2 \text{ major} + 1 = 3!$ Please rephrase this part to offer a clearer text for the reader.

271 as the type locality for *dauidii* is China, if anything, it should be the Western Pal *dauidii* that would need additional taxonomic scrutiny (contra e.g. your ref. 15). And before suggesting to use the name *bulgaricus*, a number of older ones should be evaluated, e.g. in the Caucasus (see your ref. 12 and 15).

280 It is unclear how you sorted the samples prior to analyses. Probably according to STRUCTURE clusters?

286 How many concordant and how many discordant?

308 and 312 if the introgression really happened several times independently, then this could also suggest a random lineage sorting, which would cancel your argument about that possibility earlier in the text.

318 this seems to contradict your hypothesis of an old, stable pop (l. 213).

326 along PCA3?

Fig. 4 (l. 585) What is the meaning of the "missing" category? How did you treat these "missing" genotypes in the STRUCTURE and BPP analyses?

367-369 given the continuity of nuclear genotypes between the Balkans and the Caucasus, there are more ancient names available to name this taxon, before suggesting the use of *bulgaricus*!

371 "genetically" is too vague in this context. Did you mean nuclearly?

380 High can you refer to previous studies on ecological plasticity of these two species, while you present here for the first time a new taxonomic arrangement? Anything presented so far should have confused the animals in the Balkans, giving a false idea of variation.

Author's Response to Decision Letter for (RSOS-191805.R0)

See Appendix A.

RSOS-191805.R1 (Revision)

Review form: Reviewer 1

Is the manuscript scientifically sound in its present form?

Yes

Are the interpretations and conclusions justified by the results?

Yes

Is the language acceptable?

Yes

Do you have any ethical concerns with this paper?

No

Have you any concerns about statistical analyses in this paper?

No

Recommendation?

Accept with minor revision (please list in comments)

Comments to the Author(s)

For STRUCTURE analysis, the authors still run only four replicates for each K, which are not enough to generate meaningful results. This has been suggested clearly by the second reviewer. I do not know why the authors did not do that because inconsistent results may exist in 10 replicates for some of K based on my own experience in STRUCTURE analysis. In addition, the authors did not show the results of Evanno (at least show the resulting figure in supporting information).

One minor comment in Reference:

L474, *Myotis*; L477, *mystacinus*; L538, *Myotis*, should use italics.

Review form: Reviewer 2

Is the manuscript scientifically sound in its present form?

Yes

Are the interpretations and conclusions justified by the results?

Yes

Is the language acceptable?

Yes

Do you have any ethical concerns with this paper?

No

Have you any concerns about statistical analyses in this paper?

Yes

Recommendation?

Accept with minor revision (please list in comments)

Comments to the Author(s)

This MS reports an interesting case of apparent asymmetric gene introgression from one species into the genetic background of another one. To show this, the authors use a mitochondrial marker and four nuclear introns. The mtDNA shows deep separation between two main haplogroups. Surprisingly, none of the nuclear introns show a comparable structure, as all four present allele variation that are star-like networks. It is only when information is combined (in analyses like PCA) that two, perfectly distinct groups appear. According to these two groups (corresponding to *M. davidii* and *M. mystacinus*, respectively), the authors show that most samples from the Balkans and Anatolia (but not the Caucasus, Black Sea) are pure-bred parental *M. davidii*, but carry an mtDNA copy inherited from *M. mystacinus*. Pure-bred *mystacinus* with concordant mtDNA exist in few of those areas, but nuclearly admixed individuals were not found.

I wonder whether this absence of admixed individuals result from a lack of power of their markers (too few variable sites or too much homoplasy) or really represent a total absence of contemporary hybridization (this should be discussed).

I would also know whether this total asymmetry of introgression of mtDNA could be due to an ancient pseudogene in mtDNA shared by the two species or to an effect of positive selection of haplotype-M.

Visibly, the authors did a good job in fulfilling the suggestions of two previous reviewers, but these questions remain undiscussed.

I also found the chapter 4.4 about the taxonomy a bit strange: the real *davidii* being in the Eastern clade (for both nuclear and mtDNA), these should be the nominal *M. davidii*. All the western ones (be they from the Balkans, Anatolia or the Caucasus form a single unit) should be designed (perhaps at the subspecific level) with the oldest name available in this region (*caucasicus*? I don't have references to check), not the most recent (*bulgaricus*). To my view, this chapter should either propose a full solution, or be omitted.

The MS in general is fine and besides the few remarks made before, I have only minor changes or questions to propose (line # indicated):

32 came into secondary contact during...

37 In the introgression area, *M. mystacinus* individuals with concordant nuclear and mitochondrial genotypes were ID...

64 forming overlapping distributions...

72 Species interactions during...

77 are not as reproductively isolated...

98 both taxa...

111 in suppl. Appendix, country codes must be explained in the legend.

142 Europe

143 typos and (1) and

144 genetically discordant

162 4 independent chains of 100k look really few; usually make at least 10 of 1 mio generation to get stable results; and report if you got other patterns (e.g. admixed ind.) as this would strengthen your interpretations.

197 Europe is too vague; give precise country.

205 Which clades are these? *Ikonnikovi* or *davidii* or else?

208 actually the clade M shows NO (or very weak) structure, not a strong one! Colors are mixed in your figure 2, not structured.

240 update the clade names in the figures of the appendices.

261 assignments with *davidii* nuclear background with clade M mtDNA

294 *mystacinus* (=clade M)

374 don't forget the Caucasus, which also has pure *M. mystacinus* and a few introgressed *dauidii*.

Fig. 4: introns have wrong names (ABHD and ROGDI).

Decision letter (RSOS-191805.R1)

24-Mar-2020

Dear Dr Coraman:

On behalf of the Editors, I am pleased to inform you that your Manuscript RSOS-191805.R1 entitled "Patterns of mtDNA introgression suggest population replacement in Palearctic whiskered bat species" has been accepted for publication in Royal Society Open Science subject to minor revision in accordance with the referee suggestions. Please find the referees' comments at the end of this email.

The reviewers and Subject Editor have recommended publication, but also suggest some minor revisions to your manuscript. Therefore, I invite you to respond to the comments and revise your manuscript.

- Ethics statement

- Data accessibility

<http://datadryad.org/submit?journalID=RSOS&manu=RSOS-191805.R1>

- Competing interests

- Authors' contributions

- Acknowledgements

- Funding statement

Because the schedule for publication is very tight, it is a condition of publication that you submit the revised version of your manuscript before 02-Apr-2020. Please note that the revision deadline will expire at 00.00am on this date. If you do not think you will be able to meet this date please let me know immediately.

on behalf of Dr David Ferrier (Associate Editor) and Kevin Padian (Subject Editor)
openscience@royalsociety.org

Associate Editor Comments to Author (Dr David Ferrier):

Both referees are content that this manuscript can be accepted with minor revisions, but it is essential that their recommendation to improve the STRUCTURE analysis is accommodated for this acceptance decision. Currently the STRUCTURE analysis has been done with insufficient runs and so is not really statistically valid.

The second referee also has a small number of minor suggestions for improving the discussion, which should be accommodated as far as possible.

Reviewer comments to Author:

Reviewer: 1

Comments to the Author(s)

For STRUCTURE analysis, the authors still run only four replicates for each K, which are not enough to generate meaningful results. This has been suggested clearly by the second reviewer. I do not know why the authors did not do that because inconsistent results may exist in 10 replicates for some of K based on my own experience in STRUCTURE analysis. In addition, the authors did not show the results of Evanno (at least show the resulting figure in supporting information).

One minor comment in Reference:

L474, *Myotis*; L477, *mystacinus*; L538, *Myotis*, should use italics.

Reviewer: 2

Comments to the Author(s)

This MS reports an interesting case of apparent asymmetric gene introgression from one species into the genetic background of another one. To show this, the authors use a mitochondrial marker and four nuclear introns. The mtDNA shows deep separation between two main haplogroups. Surprisingly, none of the nuclear introns show a comparable structure, as all four present allele variation that are star-like networks. It is only when information is combined (in analyses like PCA) that two, perfectly distinct groups appear. According to these two groups (corresponding to *M. davidii* and *M. mystacinus*, respectively), the authors show that most samples from the

Balkans and Anatolia (but not the Caucasus, Black Sea) are pure-bred parental *M. davidii*, but carry an mtDNA copy inherited from *M. mystacinus*. Pure-bred *mystacinus* with concordant mtDNA exist in few of those areas, but nuclear admixed individuals were not found.

I wonder whether this absence of admixed individuals result from a lack of power of their markers (too few variable sites or too much homoplasy) or really represent a total absence of contemporary hybridization (this should be discussed).

I would also know whether this total asymmetry of introgression of mtDNA could be due to an ancient pseudogene in mtDNA shared by the two species or to an effect of positive selection of haplotype-M.

Visibly, the authors did a good job in fulfilling the suggestions of two previous reviewers, but these questions remain undiscussed.

I also found the chapter 4.4 about the taxonomy a bit strange: the real *davidii* being in the Eastern clade (for both nuclear and mtDNA), these should be the nominal *M. davidii*. All the western ones (be they from the Balkans, Anatolia or the Caucasus form a single unit) should be designed (perhaps at the subspecific level) with the oldest name available in this region (*caucasicus*? I don't have references to check), not the most recent (*bulgaricus*). To my view, this chapter should either propose a full solution, or be omitted.

The MS in general is fine and besides the few remarks made before, I have only minor changes or questions to propose (line # indicated):

32 came into secondary contact during...

37 In the introgression area, *M. mystacinus* individuals with concordant nuclear and mitochondrial genotypes were ID...

64 forming overlapping distributions...

72 Species interactions during...

77 are not as reproductively isolated...

98 both taxa...

111 in suppl. Appendix, country codes must be explained in the legend.

142 Europe

143 typos and (1) and

144 genetically discordant

162 4 independent chains of 100k look really few; usually make at least 10 of 1 mio generation to get stable results; and report if you got other patterns (e.g. admixed ind.) as this would strengthen your interpretations.

197 Europe is too vague; give precise country.

205 Which clades are these? *Ikonnikovi* or *davidii* or else?

208 actually the clade M shows NO (or very weak) structure, not a strong one! Colors are mixed in your figure 2, not structured.

240 update the clade names in the figures of the appendices.

261 assignments with *davidii* nuclear background with clade M mtDNA

294 *mystacinus* (=clade M)

374 don't forget the Caucasus, which also has pure *M. mystacinus* and a few introgressed *davidii*.

Fig. 4: introns have wrong names (ABHD and ROGDI).

Author's Response to Decision Letter for (RSOS-191805.R1)

See Appendix B.

Decision letter (RSOS-191805.R2)

23-Apr-2020

Dear Dr Coraman,

It is a pleasure to accept your manuscript entitled "Patterns of mtDNA introgression suggest population replacement in Palearctic whiskered bat species" in its current form for publication in Royal Society Open Science.

Please ensure that you send to the editorial office the individual files for each figure and table included in your manuscript. You can send these in a zip folder if more convenient. Failure to provide these files may delay the processing of your proof. You may disregard this request if you have already provided these files to the editorial office.

on behalf of Dr David Ferrier (Associate Editor) and Kevin Padian (Subject Editor)
openscience@royalsociety.org

Associate Editor Comments to Author (Dr David Ferrier):

Associate Editor

Comments to the Author:

Thank you for accommodating all of the suggested revisions.

Appendix A

Editor comments:

Thank you for your submission. We recognize that this is a complex problem with many dimensions, and the reviewers have identified some areas that require further clarity or explanation. Please address these specifically as you revise, and if you need more time contact our editorial office. Best wishes.

>> We highly appreciate that you and the reviewers recognize the complexity of observed phylogeographic patterns. We find the reviewers comments highly constructive and addressed them specifically. We followed their suggestions and made the necessary changes accordingly. We changed the related figures, tables, and supplementary files. We think that these changes improved the clarity of the manuscript.

Reviewers' Comments to Author:

Reviewer: 1

Comments to the Author(s)

Comments:

Although MtDNA introgression between lineages in bats is now very common, this MS is still very interesting considering population replacement between two bat species. The logic of this MS is good. But I have some major concerns before this MS is published.

1) Numerous studies have now shown the possible occurrence of selection on mtDNA. So I wonder whether the authors have conducted positive selection analyses on ND1 which was used in their phylogeographic study? If ND1 here is not neutral, it should not be used for phylogeographic or demographic analyses.

>> We didn't perform a positive selection analysis for the ND1-fragment we analysed, which represents only a small part of the whole mitochondrial genome. ND1 is one of the most commonly used mtDNA marker in bat phylogeography and phylogenetic studies. Previous studies compared various mtDNA markers and found similar phylogenetic signals (for example Mayer et al., 2007), suggesting that it is neutral. We also selected to use this marker to utilize the previously available GenBank data.

Mayer, F., Dietz, C., & Kiefer, A. (2007). Molecular species identification boosts bat diversity. *Frontiers in Zoology*, 4(4), 1742–9994.

2) I have a little concern about ncDNA results. In the network analysis, there is no clear patterns, possibly due to the small number of polymorphic sites in these four introns. However, in the population structure and nuclear phylogeny, there are clear structuring or discordant clads. Because ncDNA results here are very important to assign admixed individuals in the contact regions of the two species, (if possible) I suggest the authors to sample more number of nuclear loci to confirm the current results.

>> We agree that increasing the number of nuclear markers will increase the resolution and will provide more insight about the evolutionary history of this species complex. Indeed, we are working on gathering genome wide data for these taxa, based on the population structure and

sympatric zones described in this study. Unfortunately, within the scope of this current manuscript, we don't have opportunities for further sequencing.

3) In each map of this MS, there is no any information of localities. If not familiar to the geography of the studied regions, it is very difficult to follow the results and discussion. Suggest to add names for some important regions in this study.

>> We add important region names into the figures.

4) In the discussion, there are too many speculations on evolutionary history of divergence and gene flow between the two studied species. I suggest the authors to conduct some formal demographic model tests (such as using Approximate Bayesian computation approach).

>> In the earlier stages of this study, we considered running Approximate Bayesian Computation analysis. However, we have only four nuclear markers, which have relatively low resolution, and mutation rates for these markers are unknown. Therefore, we decided to use Bayesian demographic approach by using relative rates and to propose possible scenarios for the observed phylogeographic patterns. In the follow-up studies, we are planning to test the scenarios that are described in this manuscript by using the whole genome dataset.

5) In the discussion, the authors mainly focus on their studied species and no general implications of this study (e.g. how species interactions can be inferred from their phylogeographic patterns and how their interactions affect their distributions) have been discussed in a wide range of animals or plants, which is important for other readers who do not study bats.

>> We added a paragraph to the end of the discussion:

“Our study demonstrates how species interactions, both in terms of gene flow and interspecific competition, play a role in the evolutionary history of related species and how these interactions can be inferred from phylogeographical patterns. Various studies identified evidence for past gene flow in diverse taxonomic groups including birds [4], cats [6], fishes [8] and humans [5]. Such genetic introgressions suggest that species interactions were widespread and likely had a crucial role in shaping the evolutionary history of species. Using a multilocus phylogeographic approach can elucidate the role of species interactions in shaping the current distribution patterns of related taxa.”

Minor comments:

1) 2.1. Genetic data sampling. Line 113: “a subset of samples”, suggest to provide specific number here.

>> We added samples sizes into the text and they are also shown in Table 2.

2) 3.5 Demographic analysis. What is the divergence rate for ncDNA used here?

>> We didn't use a divergence rate for the nuDNA and kept the expansion time estimates in “mutations/site” unit (see figure 6). Here, our main aim was to compare their relative timing.

Reviewer: 2

The authors present results of a genetic survey based on wing punches collected on whiskered bats sampled across Eastern Europe, the Middle East and Central Asia. They used one mitochondrial (ND1) and four nuclear introns to test phylogeographic patterns and discovered that these bats are represented by two major, non-coincident lineages according to these markers. The discordant pattern between the two classes of markers is located in the Balkans and in Anatolia, while animals from elsewhere in Europe and the Caucasus show complete concordance. Although none of the tested nuclear marker shows any major subdivision of alleles (all four show more or less a star-like structure of networks), when analysed together, they support the existence of two discrete clusters, corresponding each to distinct species (*M. mystacinus* and *M. davidii*, respectively). The authors interpret the areas of discordance as resulting from past events of (asymmetric) introgression leading to a replacement of the mitochondrial genome of one species into the other species. Some rough estimates of expansion time are also given, but they are quite basic (no variance around molecular rates or in estimates is taken into consideration). In the discussion the authors try to fit these results to two narrative scenarios, one of which seems more likely (post LGM expansion of one species into the range of the other).

These results are interesting as they cover a significant part the Western Palearctic distribution of those two species and include both mtDNA and nucDNA markers in their analyses. However, given the complexity of the patterns of introgression in these two classes of markers, I struggled a lot to keep track of which analysis the author were refereeing to in the text, as they used “mystacinus” or “davidii” to designate either the species themselves (but how were they identified a priori is a mystery), to refer to mitochondrial clades, or to refer to nuclear “clades” (these are indeed not clades in a cladistic sense; clusters or groups should be used instead).

>> Please see below. We followed the suggestion proposed below, which was very helpful.

The lack of clarity is blatant in the figure legends (e.g. meaning of the stars in Fig. 2b) and in the discussion, and does not help for a smooth understanding of their points.

>> Figure legend has this explanation: “Haplotypes marked with stars are shared with *M. davidii*.”

It is also unclear what they call “eastern davidii” as the sampling covered e.g. in Fig. 3 or in Fig. 5 obviously varies.

>> The distribution of eastern davidii is shown in figure 1 and figure 3. In figure 5, we focus to the introgression area. We added this explanation to the legend: “, excluding the eastern Clade D”

The same is true for the meaning of “N” mentioned in the Table 1 (=individuals analysed?) and Table 2 (no idea what it means, individuals, alleles, etc.).

>> Changed them as “number of individuals (*N*).”

Perhaps a good way to improve the clarity throughout the manuscript would be to simply name the mtDNA haplogroups “Clade 1” and “Clade 2”, restrain the names “mystacinus” and “davidii” to the two nuclear clusters, and use the term “alleles” for the nuclear variants and “haplotypes” for the mtDNA one (I know, this is sometimes also confusing in the existing literature). Finally, you could just equate these two clusters to the two species, avoiding the confusing ID issued from mtDNA vs nucDNA vs morphology.

>> We liked this idea! We renamed the mtDNA lineages as Clade M and Clade D in reference to *mystacinus* and *davidii*, which we think will be easier to follow. We change all the related tables and figures. We also went through the usage of alleles and haplotypes.

Regarding the methods, it appears that STRUCTURE is of fundamental importance in clustering the multilocus genotypes into two groups. Hence all necessary parameters need to be detailed, which is not the case. How many K were tested? Did you include for instance $K = 1$? How many replicates for each K (3 is by far insufficient; usually 10-20 are more meaningful)? It should also be mentioned and justified whether you used the admixture model and the correlated allele frequencies, or not, and whether you restricted your analyses to alleles of nuclear introns only. A good application of these methods in the context of gene introgression can be found in e.g. ESSELSTYN et al. 2013. Carving out turf in a biodiversity hotspot: multiple, previously unrecognized shrew species co-occur on Java Island, Indonesia. *Molecular Ecology* 22:4972-4987.

>> We added the necessary information. Now the section reads as:

“For these analyses, we used the allele data of phased sequences. For PCA, we used the R package adegenet 2.1.1 [30]. Samples which had data for at least three intron markers were used. For STRUCTURE analysis, we ran four independent 100,000 chains and discarding the first 20% for $K = 1-5$. We used the admixture model with the correlated allele frequencies. We repeated the runs also with the ‘loc prior’ model [31] assigning individuals into five populations, which were described in the *BEAST analysis section. These runs were merged in CLUMPAK server [32] and visualized in R.”

I see that some important results (*BEAST) are not shown but should be given at least in the suppl. info..

>> *BEAST results are shown in Figure 5d.

Besides these general considerations, I have made a number of more minor suggestions/corrections/questions near the following line numbers:

36 ...indigenous?

>> Changed.

60 ...extent rather than strength?

>> Changed.

61 ...evolved towards different ecologies while in allopatry?

>> We keep it as “ecologically differentiated during their time in isolation” to highlight the time in isolation.

75 separated or reproductively isolated?

>> Changed as separated.

95 their morphological distinction is difficult; supposed to coexist

>> Changed.

98 avoid double yet

>> Removed.

114 the four nuclear introns must be named in full here, not only abbreviated in various tables and figures.

>> Added.

136 How did you manage the phased alleles for the *BEAST and BPP analyses?

>> Explained in lines 117-118: “We used the PHASE algorithm as implemented in DNASP 5.10 [23] to phase the nuclear sequences.”

139 What about the Anatolian *mystacinus*?

>> Added the information: “The Europe (including the Anatolian *mystacinus*)”

183 divergent

>> Changed.

243 How many samples are discordant? Perhaps this important information should be highlighted in Fig. 5a with different symbols? That important part of the Fig. 5a is anyway impossible to understand, one cannot see the colour of the circles anyway!

>> We added the number of discordant individuals into the text and revise the figure 5. Now, we only use bold black borders to only highlight discordant individuals.

253 I find it strange that you did not use the second largest component, but the third for this figure. Can you justify this choice? Or does it appear in the suppl. mat.?

>> Yes, it appears in the supplementary figure. PC2 and PC3 explained similar levels of variance (2.5% and 2.4% respectively). In the context of general discussion line, we preferred to present PC3 in the main figure, as it captures the east – west split of *dauidii*.

255 Sorry I cannot understand this sentence

>> We rephrased this sentence as: “This analysis showed that the Caucasus and Mongolia harboured high diversity, and the rest of the regions, including the area of discordance, clustered as a subset within the Caucasus samples.”

263 ... lineage is further split ...

>> Changed.

264 There are now 5 clades??? 2 major + 1 = 3! Please rephrase this part to offer a clearer text for the reader.

>> Rephrased as: “The A11 analysis identified all of these clades as distinct groups.”

271 as the type locality for *dauidii* is China, if anything, it should be the Western Pal *dauidii* that would need additional taxonomic scrutiny (contra e.g. your ref. 15). And before suggesting to use the name *bulgaricus*, a number of older ones should be evaluated, e.g. in the Caucasus (see your ref. 12 and 15).

>> Good point about the western *dauidii*! Now we rephrased this sentence as: “... further studies are needed to investigate the taxonomic status of the *dauidii* lineages.” We do not suggest *bulgaricus* name. We suggest that is a synonym and try to explain why it caused a confusion.

280 It is unclear how you sorted the samples prior to analyses. Probably according to STRUCTURE clusters?

>> Yes. Rephrased as: “We divided the data into two sets based on the nuclear clusters”

286 How many concordant and how many discordant?

>> Added this sentence to the end of section 3.3: “Overall, 48 individuals had discordant mtDNA and nuclear assignments, and 78 were concordant.”

308 and 312 if the introgression really happened several times independently, then this could also suggest a random lineage sorting, which would cancel your argument about that possibility earlier in the text.

>> We agree. Based on this finding, we suggest that this scenario is unlikely: “the evidence for multiple introgression events suggest that the scenario of reproductive isolation breakdown is unlikely.”

318 this seems to contradict your hypothesis of an old, stable pop (l. 213).

>> The explanation in L. 213 refers to the mtDNA haplotypes, which introgressed to the expanding *M. dauidii*.

326 along PCA3?

>> Here we refer to the PCA for *M. dauidii*; now reads as: “In the PCA for *M. dauidii*, most of the variance was in the Caucasus populations, and the rest of the samples grouped within them, suggesting a subset of them expanded to the west (Supplementary figure 3).”

Fig. 4 (l. 585) What is the meaning of the “missing” category? How did you treat these “missing” genotypes in the STRUCTURE and BPP analyses?

>> “Missing” category refers to the individuals which did not have ND1 sequences. “Individuals which did not have mtDNA sequences are colored as grey” added to the legend. In BPP analysis,

we used a subset of samples as in the *BEAST analysis. Here we selected samples which had mtDNA identity. In the STRUCTURE analysis we didn't use mtDNA information.

367-369 given the continuity of nuclear genotypes between the Balkans and the Caucasus, there are more ancient names available to name this taxon, before suggesting the use of *bulgaricus*!

>> We agree; indeed, we do not suggest to use *bulgaricus*, but refer it as “a synonym of *M. davidii*.”

371 “genetically” is too vague in this context. Did you mean nuclearly?

>> Yes; now the sentence reads as: “that are assigned as *M. mystacinus* in nuclear markers.”

380 High can you refer to previous studies on ecological plasticity of these two species, while you present here for the first time a new taxonomic arrangement? Anything presented so far should have confused the animals in the Balkans, giving a false idea of variation.

>> Most of the studies we refer in this section are done in allopatric regions. Yet we added this explanation to the text: “(it has to be noted that these studies might have confused the species identifications, especially in the introgression area).”

Appendix B

Associate Editor Comments to Author (Dr David Ferrier):

Both referees are content that this manuscript can be accepted with minor revisions, but it is essential that their recommendation to improve the STRUCTURE analysis is accommodated for this acceptance decision. Currently the STRUCTURE analysis has been done with insufficient runs and so is not really statistically valid.

The second referee also has a small number of minor suggestions for improving the discussion, which should be accommodated as far as possible.

- We rerun the STRUCTURE analysis with 10 replicates. The results, hence our interpretations did not change. We revised the related parts in the methodology and updated the related figures in the main text and the supplementary files. We added the Evanno analysis requested by the referees to the supplementary file. We also added the discussion pointed by the second referee and implemented his/her minor suggestions. We would like to thank all the editors and the reviewers for their constructive comments.

Reviewer comments to Author:

Reviewer: 1

Comments to the Author(s)

For STRUCTURE analysis, the authors still run only four replicates for each K, which are not enough to generate meaningful results. This has been suggested clearly by the second reviewer. I do not know why the authors did not do that because inconsistent results may exist in 10 replicates for some of K based on my own experience in STRUCTURE analysis. In addition, the authors did not show the results of Evanno (at least show the resulting figure in supporting information).

- We rerun the STRUCTURE analysis with 10 replicates and revised the related parts in the text and the figures. The results were concordant with the 4 replicates. We also added the results of the Evanno to the supplementary file.

One minor comment in Reference:

L474, *Myotis*; L477, *mystacinus*; L538, *Myotis*, should use italics.

- Corrected

Reviewer: 2

Comments to the Author(s)

This MS reports an interesting case of apparent asymmetric gene introgression from one species into the genetic background of another one. To show this, the authors use a mitochondrial marker and four nuclear introns. The mtDNA shows deep separation between two main haplogroups. Surprisingly, none of the nuclear introns show a comparable structure, as all four present allele variation that are star-like networks. It is only when information is combined (in analyses like PCA) that two, perfectly distinct groups appear. According to these two groups (corresponding to *M. davidii* and *M. mystacinus*, respectively), the authors show that most samples from the Balkans and Anatolia (but not the Caucasus, Black Sea) are pure-bred parental *M. davidii*, but carry an mtDNA copy inherited from *M. mystacinus*. Pure-bred *mystacinus* with concordant mtDNA exist in few of those areas, but nuclearly admixed individual were not found.

I wonder whether this absence of admixed individuals result from a lack of power of their markers (too few

variable sites or too much homoplasy) or really represent a total absence of contemporary hybridization (this should be discussed).

- We added this explanation: “We did not identify any admixed individuals for $K = 2$. This might represent a lack of contemporary hybridization between these groups, but also might be affected by the resolution of analyzed nuclear markers. Further studies utilizing genomic approaches can provide more definitive results.”

I would also know whether this total asymmetry of introgression of mtDNA could be due to an ancient pseudogene in mtDNA shared by the two species or to an effect of positive selection of haplotype-M.

- We think that if it would be a pseudo gene, then we would expect to find it also in the eastern ranges as well, not geographically restricted. However, this (introgressed) mitochondrial lineage in *dauidii* occurs where both species have overlapping ranges. Furthermore, the mtDNA of both species in same areas are closer to each other than their conspecifics in other regions, which indicates an introgression pattern.

About the positive selection, we cannot test it with our data-set. But the patterns of genetic structure in the Caucasus suggest that both species can occur with both of the mitochondrial lineages. In that case, we would expect them to have *mystacinus* mtDNA only as well. We can possibly answer these questions with a genomic approach.

Visibly, the authors did a good job in fulfilling the suggestions of two previous reviewers, but these questions remain undiscussed.

I also found the chapter 4.4 about the taxonomy a bit strange: the real *dauidii* being in the Eastern clade (for both nuclear and mtDNA), these should be the nominal *M. dauidii*. All the western ones (be they from the Balkans, Anatolia or the Caucasus form a single unit) should be designed (perhaps at the subspecific level) with the oldest name available in this region (*caucasicus*? I don't have references to check), not the most recent (*bulgaricus*). To my view, this chapter should either propose a full solution, or be omitted.

- In this section, our aim was to address the taxonomic status of *bulgaricus*, which is a long debated taxon. For that taxon, we propose a full solution. About the possible distinction between eastern and western subclades of Clade D, further work is needed. We think this study will stimulate such studies. We added this sentence to the end of this paragraph: “In case, further studies would identify that the subclades of Clade D are distinctive at a taxonomic level, then the name *aurascens* should be recalled to name the western Clade D.”

The MS in general is fine and besides the few remarks made before, I have only minor changes or questions to propose (line # indicated):

32 came into secondary contact during...

- Added secondary

37 In the introgression area, *M. mystacinus* individuals with concordant nuclear and mitochondrial genotypes were ID...

- Added.

64 forming overlapping distributions...

- Changed

72 Species interactions during...

- Changed.

77 are not as reproductively isolated...

- Changed.

98 both taxa...

- Changed.

111 in suppl. Appendix, country codes must be explained in the legend.

- Added the explanation.

142 Europe

- We keep “the” as it refers to the clade.

143 typos and (1) and

- Corrected

144 genetically discordant

- Added

162 4 independent chains of 100k look really few; usually make at least 10 of 1 mio generation to get stable results; and report if you got other patterns (e.g. admixed ind.) as this would strengthen your interpretations.

- Increased the number of repeats to 10. The length of the runs is sufficient as the values quickly diverge with relatively few markers.

197 Europe is too vague; give precise country.

- Changed as “Greece.”

205 Which clades are these? Ikonnikovi or davidii or else?

- We refer to the subclades that are described in the previous sentence. Changed it as “these subclades.”

208 actually the clade M shows NO (or very weak) structure, not a strong one! Colors are mixed in your figure 2, not structured.

- Changed as weak.

240 update the clade names in the figures of the appendices.

- Updated.

261 assignments with davidii nuclear background with clade M mtDNA

- Added.

294 mystacinus (=clade M)

- Changed

374 don't forget the Caucasus, which also has pure *M. mystacinus* and a few introgressed *dauidii*.

- True. In this section, we discuss about the population replacement. As both species occur in the Caucasus with some level of introgression, we only refer to the Balkans and Anatolia in the subsection header.

Fig. 4: introns have wrong names (ABHD and ROGDI).

- Corrected